# Multi-Scale Characterization of Spatial Variability of Soil Organic Carbon in a Semiarid Zone in Northern China

**Long Li** [1,2,3]**, Yongjie Yue** [4,]*****, Fucang Qin** [1,2,3,4]**, Xiaoyu Dong** [1]**, Cheng Sun** [1]**, Yanqi Liu** [1] **and Peng Zhang** [1]

1   College of Desert Control Science and Engineering, Inner Mongolia Agricultural University, Hohhot 010018, China; lilongdhr@126.com (L.L.); 15034777976@126.com (F.Q.); dongxiaoyu2019@126.com (X.D.); suncheng0206@126.com (C.S.); liuyanqi93@126.com (Y.L.); zp54152022@163.com (P.Z.)
2   Key Laboratory of Desert Ecosystem Conservation and Restoration, State Forestry and Grassland Administration of China, Hohhot 010018, China
3   Key Laboratory of Aeolian Physics and Desertification Control Engineering from Inner Mongolia Autonomous Region, Hohhot 010018, China
4   Forestry College, Inner Mongolia Agricultural University, Hohhot 010018, China
*   Correspondence: wolongyue@126.com

**Abstract:** The variation of soil organic carbon (SOC) spatial distribution is dependent on the relative contributions of different environmental factors, and the dominant factors change according to study scales. Here, geostatistical and remote sensing techniques were used to gain deep knowledge about SOC spatial distribution patterns and their dominant determinants at different study scales; specifically, the structure of the spatial variability of SOC content at the county, regional, and watershed scales in Aohan, China were analyzed. The results show that altitude and normalized difference vegetation index (NDVI) are the key predictors explaining 49.6% of the SOC variability at the county scale; NDVI and slope are the key predictors explaining 36.2% of the SOC variability at the regional scale; and terrain factors are the most significant factors at the watershed scale. These three scales have a moderate spatial correlation in terms of SOC content. As the study scale widens, the spatial variability attributable to the random factors increases gradually, whereas the variability attributable to the structural factors gradually weakens. Soil type and land use type are the key factors influencing the SOC content at these three scales. At all scales, the SOC contents of the different land use types differ significantly in the order forestland > shrubland > grassland. Conservation of regional soil and water and prevention of soil desertification are effective measures for improving SOC content.

**Keywords:** multi-scale; soil organic carbon; spatial variability; geostatistics

## 1. Introduction

The soil carbon pool storage in the global terrestrial ecosystem is estimated at 1200–2500 Pg (1 Pg = $10^{15}$ g), accounting for 75% of the total terrestrial carbon pool and is twice as large as the atmospheric carbon pool [1]. The soil carbon pool has a far-reaching influence on the composition, structure, and function of the entire terrestrial ecosystem. Even a slight change in the terrestrial carbon pool will cause a considerable impact on terrestrial ecosystems [2]. Semiarid zones, despite their important role in terrestrial ecosystems, are expanding due to global warming. Moreover, the soil organic carbon (SOC) of semiarid zones plays an important role in soil carbon pooling in terrestrial ecosystems. Thus, the spatial variability characteristics of SOC in semiarid zones, especially their distribution characteristics at different scales, should be analyzed.

The formation and evolution of soil are extremely complex, and both natural and human factors have a marked effect on soil [3]. Regardless of the scale, spatial variability in soil properties always exists, and the SOC spatial distribution patterns are closely related to the environmental factors for certain scales. Different environmental factors directly or

indirectly affect the accumulation and decomposition of SOC [4]. The spatial variability of SOC often results in an overlaid effect in the spatial distribution patterns at different scales, and the dominating factors are also altered due to the change in study scales. Incidentally, information about the SOC of a large area is difficult to determine, and soil carbon varies considerably even in the same study zone. Determining the SOC spatial variability at multiple scales is essential in obtaining a clear relation between soil and the different influencing factors [5].

Soil properties have high spatiotemporal variability and are affected by the extrinsic factors and inherent heterogeneity of soil at different scales [6,7]. Studies about the operating and interacting factors have been conducted for many years [8]. Furthermore, the relationship between SOC and its controlling factors varies greatly across different scales [9]. The dominant processes at one study scale may not have the same important impact at other scales. For example, the expected differences in SOC caused by large-scale processes, such as soil, vegetation, or topography differences, may be masked by local-scale processes, such as variations attributable to soil intrinsic properties. However, most studies on SOC spatial variations have focused on a single scale; the multi-scale comparison of SOC spatial variabilities is seldom reported. The paucity of data regarding the multi-scale characterization of SOC spatial variability has been identified as an important research field in soil science [10,11]. Simultaneous actions of different natural processes often give rise to nonlinear distributions at different scales, but the SOC distribution at these varying scales is difficult to approximate [12]. Therefore, the direct and indirect contributions of environmental drivers on SOC must also be specified at specific scales.

This study analyzed the SOC content spatial variability at the county, regional, and watershed scales and attempted to predict the multi-scale spatial distribution patterns of SOC content in Aohan County, China. The objectives of this study are to (1) gain deep knowledge of the SOC spatial variability at different scales and (2) clarify the relationship between SOC and its different influencing factors and identify the dominant factor at each scale.

## 2. Materials and Methods

### 2.1. Study Site

The study site is Aohan County (41°42′–42°02′ N, 119°30′–120°54′ E), which is located in Inner Mongolia, North China. The study area is 8300 km$^2$ in size and belongs to the region with a semiarid temperate continental climate. The temperature in the study area is in the range of −30.9 °C to 39.7 °C, the mean annual atmospheric temperature is 6 °C, and the precipitation is 310–460 mm. The mean annual precipitation decreases gradually from south to north and mainly falls between June and August. The elevation is in the range of 300 to 1250 m (Figure 1). Four soil types are distributed in the study area. From south to north, they are brown soil, cinnamon soil, chestnut soil, and aeolian sandy soil (Figure 2). Aeolian sandy soil is found in the north, reaching the southern fringe of the Keerqin sandy land, and native vegetation is composed of sand vegetation. Cinnamon soil and chestnut soil are located in the central section and mainly covered by grasses. Brown soil is mainly distributed in the south.

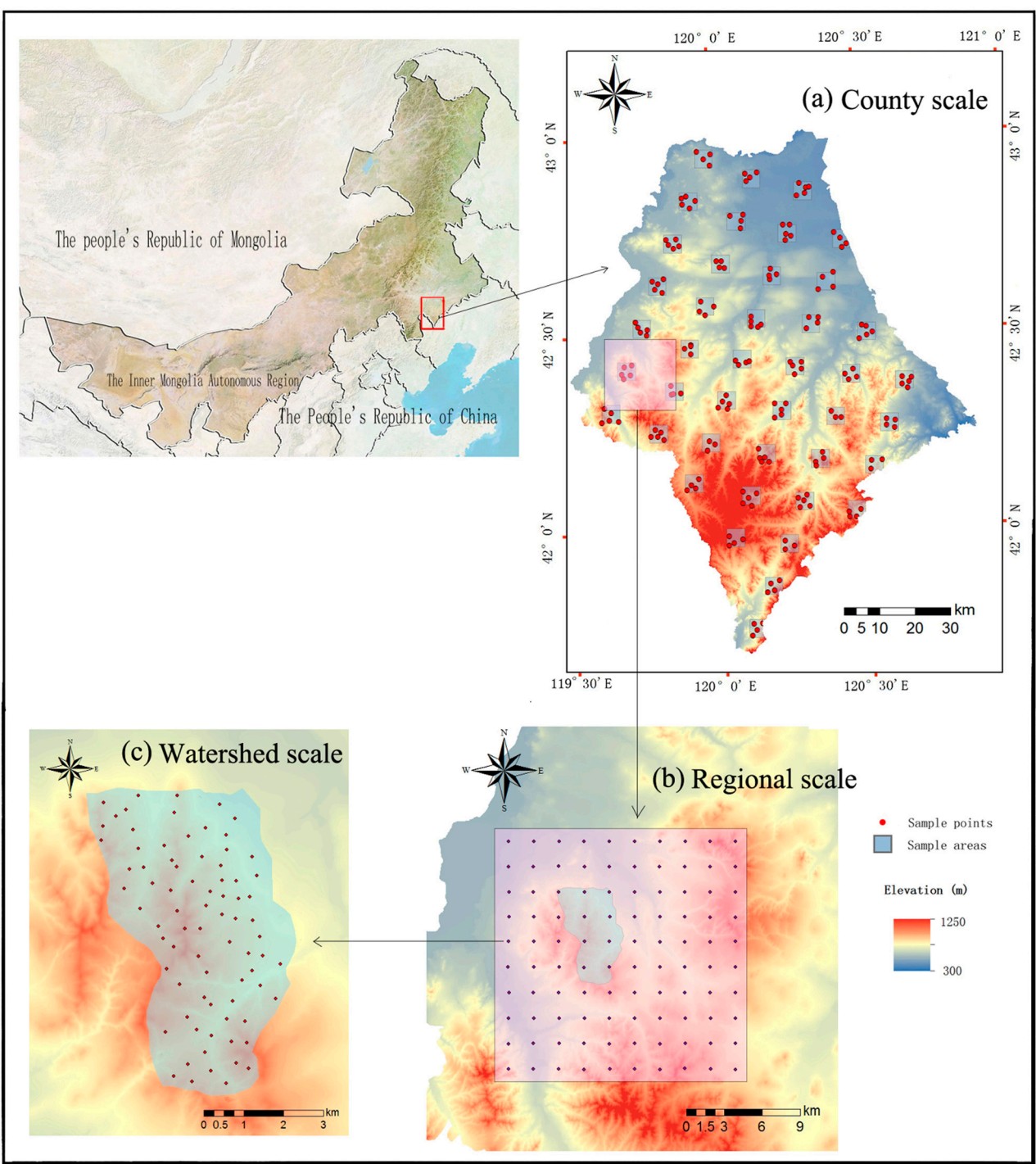

**Figure 1.** Study area and sampling sites.

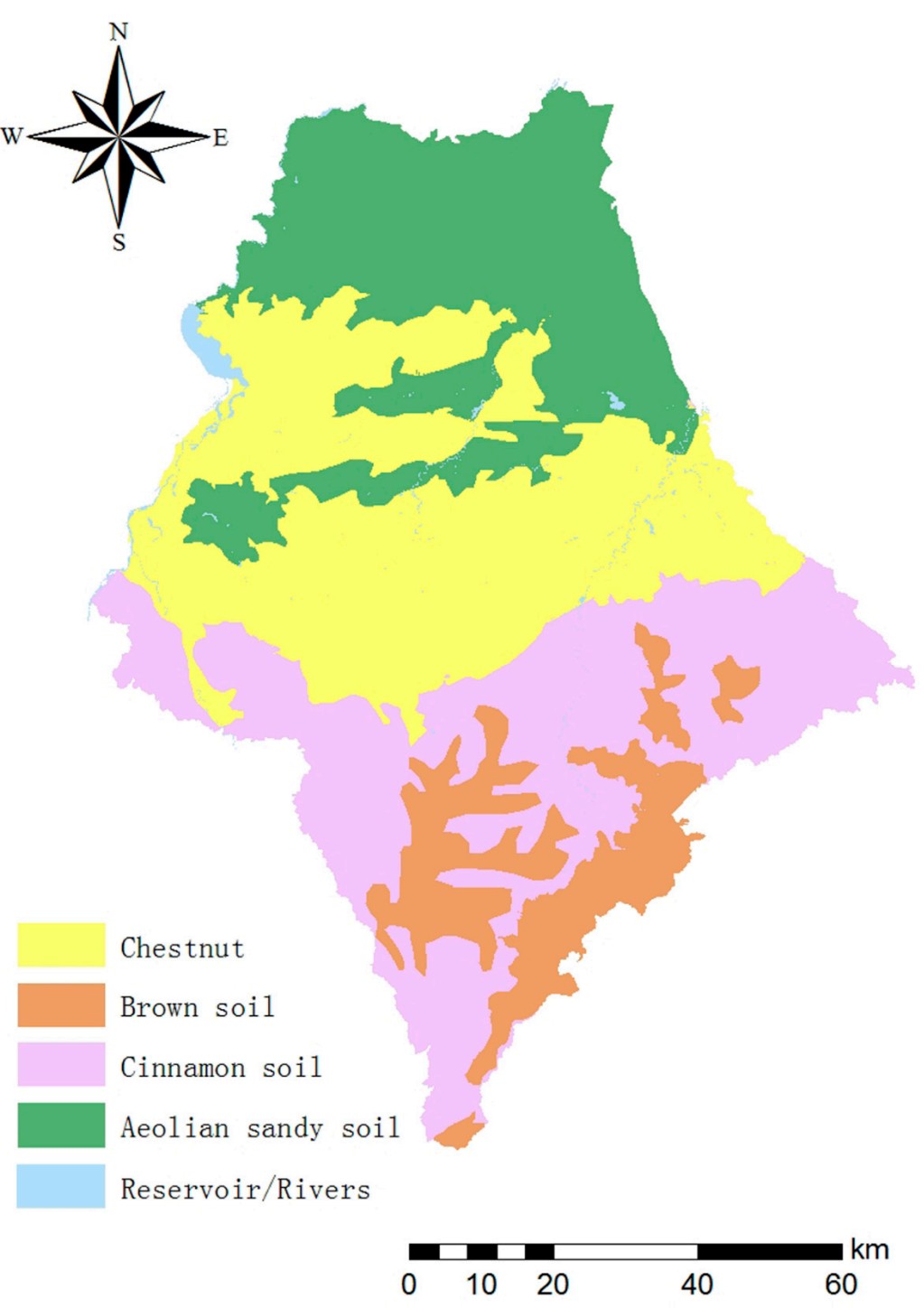

**Figure 2.** Map of soil types.

*2.2. Soil Sampling*

A total of 369 soil samples were collected in August 2021 (Figure 1). On the basis of remote sensing imagery (taken on 15 August 2020 in 30 m resolution) and the distribution maps of soil types, the sampling points were determined by combining remote sensing technology and field surveys.

2.2.1. County-Scale Sampling

The sampling system included sampling points nested in specific areas and belt transects built via large-scale sampling. Representative samples were selected to accurately

show the natural features determined via the sampling system adopted in this study. This sampling method is more efficient than the traditional approach. All soil samples were collected using a small soil drill, and the distance between each sample location was 3 m.

(1) Belt transects: Six belt transects were arranged along the northeast–southwest direction in the study area. The bandwidth between every two adjacent belt transects was 15 km.

(2) Sampling areas: Forty-two sampling areas were designed for the belt transects. The dimension of each sample grid was a 5 km × 5 km square, and the distance between two adjacent sampling areas in the same belt transect was 13 km.

(3) Sampling points: The design of each sampling point allowed for a comprehensive investigation of the influence of soil types, typical vegetation types, land use types, and topographical factors on SOC. Three to five representative sampling points were designed randomly in each sampling area. Thus, a total of 182 sampling points were selected.

### 2.2.2. Regional Scale Sampling

A total of 100 soil samples were collected based on a 2 km × 2 km grid-point system in a 20 km × 20 km square area in the western part of the study area. Surface soil samples were collected from the 0.00–0.20 m horizon.

### 2.2.3. Watershed-Scale Sampling

Surface soil samples were collected from the 0.00–0.20 m horizon at randomly selected points in the 27 $km^2$ watershed named Huanghuadianzi. Thus, a total of 87 sampling points were selected.

### 2.2.4. Sampling Treatment and Data Source

After determining the sampling points, surface soil samples along the 0.00–0.20 m horizon were collected, air-dried, crushed, and passed through a sieve with a 2 mm mesh. The visible plant residues called litters (dead branches and leaves) found in the soil samples were removed prior to the measurement of SOC content because plant residues were excluded from the SOC analyses. Hence, sieved soil was the only material used to determine the SOC. The SOC content was measured via dichromate oxidation and titration with ferrous ammonium sulfate [13].

The topographic map was scanned and digitized to build the geocoded elevation data in 1:10,000 scale. A raster digital elevation model (DEM) with a square cell size of 10 $m^2$ was built from the elevation data using the ArcGIS software. Data on elevation, slope, and slope aspect features were derived from the DEM. The normalized difference vegetation index (NDVI) maps obtained from Landsat TM imagery (15 August 2016, 30 m resolution) with the support of Erdas were used to characterize the vegetation types. The general form of NDVI is given by:

$$NDVI = (LNIR - LR)/(LNIR + LR), \tag{1}$$

where *LR* is the red band reflectance value, and *LNIR* is the near-infrared band reflectance value. NDVI can reflect the background influence of the plant canopy, such as soil, wetland, snow, dead leaves, roughness, and is related to vegetation coverage.

### 2.3. Statistical Analyses
### 2.3.1. Geostatistics

Geostatistical semivariogram was used to analyze the spatial variability and spatial autocorrelation of SOC in this study. In geostatistical spatial autocorrelation, the semi-variogram generally increases with distance between neighboring samples, indicating correlation, and then stabilizes at the sill ($C_0$ + C), indicating that the samples beyond this distance are spatially independent. Distance represents the spatial correlation range (*a*), and its size reflects the spatial autocorrelation scale of the regional variables. A variance

on a scale that is smaller than the field samples has a zero-lag distance, which is called the nugget effect ($C_0$). C is the structural variance representing the range of variance caused by the spatial autocorrelation in the survey data, and $C_0$ represents the experimental error, which is less than the actual sampling scale variation caused by the random part, namely, the spatial heterogeneity [14]. The ratio value of the nugget and sill (i.e., $C_0/C + C_0$) can explain both the system variability and the random part. Furthermore, ($C_0/C + C_0$) indicates the spatial variability caused by the random part, which accounts for the proportion of the total system variance. Values of <0.25, 0.25–0.75, and >0.75 represent the weak, moderate, and strong spatial variations in the SOC, respectively [14].

Here, a semivariogram was used to represent the mathematical expectation of the square of the regional variable's $Z(xi)$ and $Z(xi + h)$ increment, i.e., the variance of the regional variable. Its general form is given by:

$$r(h) = \frac{1}{2N(h)} \sum_{i=1}^{N(h)} [Z(x_i) - Z(x_i - h)]^2 \qquad (2)$$

where $Z(x_i)$ and $Z(x_i + h)$ are the measured values for the experimental data at location $x_i$ and $x_i + h$, $r(h)$ is the variogram for the lag distance $h$, and $N(h)$ is the number of data pairs separated by $h$.

Ordinary kriging was selected as the most suitable interpolation method for estimating the SOC at the non-sampled locations based on the semivariogram results. Ordinary kriging assumes a constant but unknown mean, which is unbiased for expected values of random variables and estimators, and it minimizes the variance of estimation errors [15]. Furthermore, kriging not only can determine an estimate but also gives a variance of estimation errors for quantifying the uncertainty of the estimate at each site. A spatial interpolation estimator $Z(x_0)$ was used to find the best linear unbiased estimate (at a non-sampled location) of a second-order stationary random field with an unknown constant mean. $Z(x_0)$ is given by:

$$Z(x_0) = \sum_{i=1}^{n} \lambda_i Z(x_i) \qquad (3)$$

where $Z(x_0)$ is the kriging estimate at the non-sampled location $x_0$, $Z(x_i)$ is the sampled value at location $x_i$, and $\lambda_i$ is the weighting factor for $Z(x_i)$.

Prior to analysis, the data were examined for normality by the Kolmogorov–Smirnov test. Data that were not distributed normally were log-transformed. GS+ 7.0 was used for the semivariogram analysis [16]. The prediction maps of SOC content at the different scales were created with ArcGIS (Figure 3). Moreover, all prediction maps presenting the SOC content were tested via cross validation.

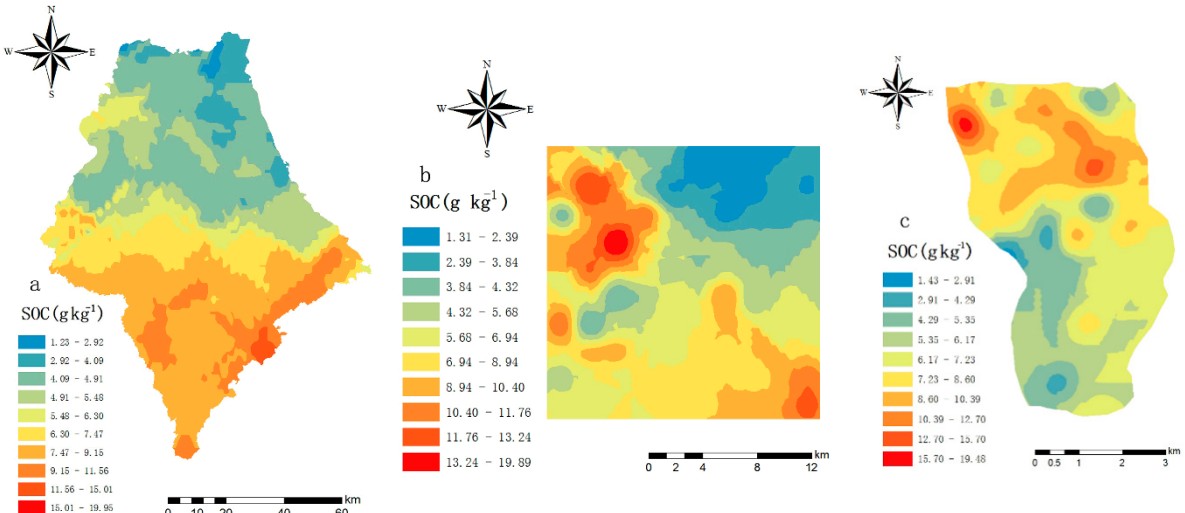

**Figure 3.** Spatial distribution of SOC at the (**a**) county, (**b**) regional, and (**c**) watershed scales.

### 2.3.2. Other Analyses

A Tukey test was used to compare the differences in SOC contents of the different land use types (forestland, shrubland, and grassland) and soil types (aeolian sandy soil, chestnut soil, cinnamon soil, and brown soil). In addition, the relationships between SOC content and control factors (altitude, NDVI, slope, and slope aspect) were examined via correlation analysis. The coefficient of variation (CV) was determined as the most discriminating factor. When the CV of SOC was <0.10, it showed a low variability; when the CV of SOC was >0.90, it showed a high variability [17].

The effects of the environmental factors on SOC at the different scales were analyzed via stepwise multi-regression. In stepwise regression, predictive variables are entered into the regression equation one at a time according to the statistical criteria. At each step of the analysis, predictive variables contribute most to the predictive equation in terms of adding multiple correlations. The process continues only when other variables add any statistics to the regression equation; otherwise, the analysis stops. After adding a new variable, it is checked whether some variables can be removed without significantly increasing the residual sum of squares (RSS) and reducing the determination coefficient ($R^2$). Therefore, in stepwise regression, not all predictive variables can be integrated into the equation. In this study, topography and vegetation were selected as the major predictor variables. Slope, slope aspect, and altitude were selected to explain the effects of topography, while NDVI was selected to characterize vegetation. All the data were analyzed using the R 3.0.1 software from AT&T's Bell Labs.

### 3. Results

*3.1. Descriptive Statistical Analysis of SOC Content*

The SOC contents at the different scales are presented in Table 1. The mean content of SOC at the county, regional, and watershed scales were 7.49, 7.57, and 7.54 g kg$^{-1}$, respectively. The frequency distributions of SOC at the three scales were all near-normal and with close skewness and kurtosis. For all scales, the frequency distribution was skewed to the right and characterized by negative kurtosis. The frequency distributions of SOC at the watershed scale were more skewed to the right compared with the other two scales. The CV analysis also showed that the SOC contents have a moderate discrete degree at the three scales. The county scale had the maximum range of SOC content, and the CV value at the county scale was higher than those of the other two scales. Moreover, the maximum value of SOC content at the county scale was greater than those of the other scales, and its minimum value was lower than those of the other scales.

**Table 1.** Descriptive statistics of SOC content at different scales in the study area.

| Scale | SOC Content (g kg$^{-1}$) | | | SD | Sample Number | CV (%) | Skewness | Kurtosis |
|---|---|---|---|---|---|---|---|---|
| | Minimum | Mean | Maximum | | | | | |
| County scale | 1.23 | 7.49 | 19.95 | 3.80 | 182 | 50.73 | 0.17 | 2.09 |
| Regional scale | 1.31 | 7.57 | 19.89 | 3.31 | 100 | 43.59 | 0.35 | 2.79 |
| Watershed scale | 1.43 | 7.54 | 19.48 | 3.50 | 87 | 46.45 | 0.86 | 2.28 |

### 3.2. Analyses of Spatial Variability

The fitting precision of different models were compared, and the theoretical models of the semivariogram for SOC content were explored (Figure 4). The spherical model provided the best fit for SOC content at county and regional scales, whereas the Gaussian model provided the best fit for the SOC content at the watershed scale. The $C_0/(C + C_0)$ of SOC for the three scales were between 27.66% and 36.61% (Table 2). Thus, a moderate spatial correlation of SOC content exists among the three scales, with the highest correlation at the county scale. The structural and random factors both influence the spatial variability of SOC. The proportions of spatial variability caused by the random part were 36.61%, 33.42%, and 27.66%. With the widening of the study scale, the spatial variability caused by the random factors increased gradually, and the structural factors were gradually weakened.

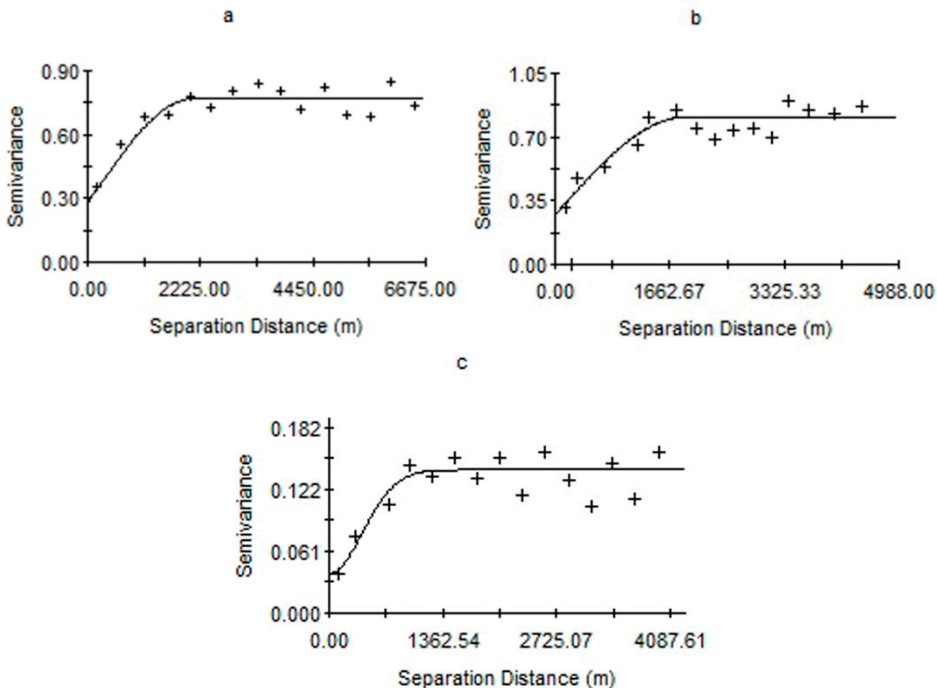

**Figure 4.** Experimental variogram and fitted model of SOC at the (**a**) county, (**b**) regional, and (**c**) watershed scales.

**Table 2.** Theory model and parameters of the semivariogram for SOC content at different scales.

| Scale | Model | Nugget | Sill | Range (m) | Nugget/Sill (%) | RSS | R$^2$ |
|---|---|---|---|---|---|---|---|
| County scale | Spherical | 0.28 | 0.77 | 2100 | 36.61 | 2.11 | 0.73 |
| Regional scale | Spherical | 0.27 | 0.81 | 1890 | 33.42 | 1.96 | 0.77 |
| Watershed scale | Gaussian | 0.039 | 0.141 | 980 | 27.66 | 2.43 | 0.50 |

Range is an important parameter in the statistical analysis of the theoretical model. Its size reflects the regional variable scale or spatial autocorrelation scale of the SOC. Here, the ranges of SOC content at the three scales were 980, 1890, and 2100 m, and the values increased with the widening of the study scale. This trend indicates that a larger

area still had a wider spatial correlation in the comparison of regional and watershed scales. In theory, the variation scale of SOC can be determined by the combined action of environmental processes.

### 3.3. Spatial Distribution of SOC Content

Ordinary kriging interpolation was used to simulate the spatial distribution patterns of the SOC at multiple scales. The results indicate that the cross-validation accuracies were 84%, 79%, and 75% at the county, regional, and watershed scales, respectively. The distribution pattern had a strong variability in a north–south trend at the county scale (Figure 3a) and showed patchy transitions in high-to-low distribution. A close correlation was observed between organic matter and distribution of soil types at the county scale. The soil types from south to north included brown soil, cinnamon soil, chestnut soil, and aeolian sandy soil. The differences in soil structures and textures were obvious among the soil types, and they directly led to changes in the decomposition and transformation efficiency of soil organic matter and the difference in SOC in terms of horizontal distribution between north and south.

The distribution patterns of SOC content showed an island-like pattern at the regional scale (Figure 3b). The low value zones of SOC content mainly appeared in the northeast, whereas the SOC contents in the northwest and southeast were high. The dependence of SOC at the regional scale may be related to the pedological component (county scale) with a topographic component (watershed scale). The differences in both soil and vegetative distribution affect the spatial variability of SOC.

The distribution of SOC content at the watershed scale showed a scattered island-like pattern (Figure 3c). This finding indicates that the high SOC is distributed in the northwestern part of the watershed with more forests, whereas the low SOC is distributed in the southwestern part within a centralized area with mostly steep slopes. The complex topography affects the distribution pattern of vegetation and soil erosion. This finding can be explained by the watershed-scale variability being controlled by plant distribution and topography.

### 3.4. Soil Type, Land Use, and Topography Impacts on SOC Content at Different Scales

The spatial distribution patterns of SOC content were affected by topographical factors, soil types, vegetation distribution, and other factors. Significant differences in SOC content were found at the three scales according to land use type (Table 3). The SOC content of forestland was significantly higher than that of shrubland and grassland at the county scale ($p < 0.05$). No significant difference in SOC content was found between forestland and shrubland at the regional and watershed scales. The SOC of grassland was significantly lower than that of shrubland and forestland at each scale ($p < 0.05$) with a low CV. Regarding the difference between forestland and grassland, the results show that SOC can decrease when forestland and shrubland are converted into grassland.

**Table 3.** Spatial distribution of SOC contents in different soil and land use types.

| | | Scale | | | | | | | | | | | |
| | | County Scale | | | | Regional Scale | | | | Watershed Scale | | | |
| | | Number of Samples | Mean (g kg$^{-1}$) | SD | CV (%) | Number of Samples | Mean (g kg$^{-1}$) | SD | CV (%) | Number of Samples | Mean (g kg$^{-1}$) | SD | CV (%) |
|---|---|---|---|---|---|---|---|---|---|---|---|---|---|
| Land use types | Forestland | 89 | 8.15 [a] | 3.61 | 44.31 | 31 | 7.61 [a] | 2.41 | 31.71 | 35 | 7.96 [a] | 2.64 | 33.13 |
| | Shrubland | 56 | 7.21 [b] | 2.79 | 38.70 | 46 | 7.15 [a] | 2.56 | 35.83 | 32 | 7.67 [a] | 2.96 | 38.64 |
| | Grassland | 37 | 4.70 [c] | 1.94 | 41.24 | 23 | 6.04 [b] | 1.78 | 29.42 | 20 | 6.67 [b] | 2.18 | 32.72 |
| Soil types | Aeolian sandy soil | 45 | 4.88 [a] | 1.34 | 27.46 | 31 | 4.96 [a] | 1.13 | 22.76 | 24 | 5.25 [a] | 1.30 | 24.79 |
| | Chestnut soil | 64 | 7.59 [b] | 3.28 | 43.21 | 55 | 6.54 [b] | 2.53 | 38.61 | 63 | 6.43 [b] | 2.34 | 36.45 |
| | Cinnamon soil | 40 | 8.25 [b] | 2.69 | 32.61 | 14 | 8.44 [c] | 2.80 | 33.23 | | | | |
| | Brown soil | 33 | 12.84 [c] | 5.05 | 39.33 | | | | | | | | |

Different letters indicate significant differences at each scale between soil types and land use types at $p < 0.05$ according to the Tukey test.

Significant differences in SOC content were found at the three scales according to soil types (Table 3). The SOC contents the in aeolian sandy soil were significantly lower than those in the other soil types at each scale ($p < 0.05$). The SOC content in brown soil showed the highest value and was nearly three times higher than that in aeolian sandy soil ($p < 0.05$). The SOC content in cinnamon soil was slightly higher than that in chestnut soil, with no significant difference at the county scale but was significantly higher at the regional scale.

*3.5. Relationship between SOC Content and Different Environmental Factors at Different Scales*

The results of the Pearson's correlation analysis between SOC content and the different environmental factors (Table 4) indicate that SOC content is highly and positively correlated with NDVI at the county and regional scales, with correlation coefficients of 0.6928 and 0.5112, respectively, and the values are significant at the 0.05 level. The SOC content is highly correlated with altitude and slope at the watershed scale.

**Table 4.** Pearson's correlations coefficients between SOC content and environmental factors at different scales.

|  | **Altitude** | **Slope** | **Slope Aspect** | **NDVI** |
|---|---|---|---|---|
| County scale |  |  |  |  |
| SOC content | 0.4522 * | 0.3376 | 0.0389 | 0.6928 * |
| Regional scale |  |  |  |  |
| SOC content | 0.3532 | −0.4123 * | 0.2345 | 0.5112 * |
| Watershed scale |  |  |  |  |
| SOC content | 0.5123 * | −0.4433 * | 0.2921 | 0.3005 |

\* Correlation is significant at $p < 0.05$.

The relationships between SOC content and each of the predictor variables (altitude, slope, slope aspect, and NDVI) were examined via stepwise multi-regression analysis. The analysis was accomplished at the county, regional, and watershed scales. Significant differences were observed in the relative importance of the predictive variables (Table 5). Altitude and NDVI were the key predictor variables, explaining 49.6% of SOC content variability at the county scale. Approximately 36.2% of the SOC content variability was attributed to NDVI and slope at the regional scale. The topographic factors, such as altitude, slope, and slope aspect, were the most significant factors controlling SOC content at the watershed scale. Therefore, vegetation factors play a more important role in determining SOC content at a larger scale.

**Table 5.** Stepwise multivariate regression model of SOC content against the predictor variables at different scales.

| Scale | Predictive Variables | Coefficients | *p* Values | Standard Error (SE) | $R^2$ Adj. | *p* Value |
|---|---|---|---|---|---|---|
| | Intercept | 0.578 | <0.001 | 0.072 | | |
| County scale | Altitude | 0.015 | <0.001 | 116.210 | 0.496 | <0.001 |
| | NDVI | 0.021 | <0.001 | 0.020 | | |
| | Intercept | 7.929 | 0.051 | 0.851 | | |
| Regional scale | NDVI | 6.069 | 0.062 | 0.094 | 0.362 | 0.034 |
| | Slope | −0.312 | 0.105 | 1.740 | | |
| | Intercept | 0.623 | 0.044 | 0.067 | | |
| Watershed scale | Altitude | 0.045 | 0.023 | 47.470 | 0.457 | 0.019 |
| | Slope | −0.234 | 0.045 | 13.210 | | |
| | Slope aspect | 0.012 | 0.084 | 0.640 | | |

## 4. Discussion

### 4.1. Multi-Scale Characterization of SOC Spatial Variability

The results of this study suggest that soil types and land use types have large effects on the spatial variability of SOC at different scales (Table 3), which can explain why the SOC distribution has patchy and island-like patterns (Figure 3). This finding can be attributed to complex factors that break the homogenous distribution of SOC at the larger scale [18], which provides the special spatial environment for the distribution difference; the outliers are more apparent in extreme environments. The spatial variability derived from the structural factors may be related to the topographical and climatic factors and the soil parent materials [19]. Previously, Cambardella et al. [20] found that human activities at local scales weaken the spatial correlation of natural soils, and many kinds of complex features from the external environment can enhance the SOC spatial variability at the large scale. Meanwhile, the county scale variation can reflect the changes in land use types and parent materials, both of which strongly influence the basic pedological properties [21].

The presence of trees or shrubs on originally grassed areas can, after decades, significantly improve the near-surface SOC, resulting in a patchy spatial distribution of soil organic matter [22,23]. Previously, Post and Kwon [24] demonstrated that changes in land use types lead to differences in SOC content. The uncertainty about the magnitude of these changes at different scales is still prevalent. Here, the SOC content in the grassland was significantly lower than that in the forestland and shrubland, and the SOC content in the forestland was the highest, with significant differences at the county scale ($p < 0.05$). Therefore, an important consideration concerning the reduction of SOC content is the conversion of forestland into grassland. The conversion from forestland to grassland results in the reduction of biomass inputs into the soil and the acceleration of the decomposition of organic matter, leading to reductions in surface SOC content [25–27]. Juan et al. [3] found similar SOC losses when forestland was converted into shrubland or cropland, as evidenced by the 0–40 cm of soil at the provincial scale in a semi-arid region in Spain. Thus, changing the study scales has no significant correlation with the effects of land use types on SOC. Other major sources of SOC are the decomposition and supplementation of vegetation litter. Among the different vegetation types, the root distribution patterns and decomposition levels of litter influence the inputs into SOC [19]. Primary productivity is higher in forestland with rich biomass. Therefore, soil can obtain more abundant SOC sources from forestland than the other land use types at any scale.

Soil type is another important factor affecting the spatial variability of SOC at different scales. The soil parent material plays an important role in the organic matter stored in soil [28]. This study determined that the SOC content in aeolian sandy was significantly lower than those in other soils (Table 3). Forstner et al. [29] also found that aeolian sandy soil with low soil fertility seldom contains SOC. The main goal of ecological construction is to achieve sand fixation in aeolian sandy soil due to the long-term influence of wind erosion as soil erosion removes the surface litter particles attached to organic matter which further reduces SOC accumulation [30]. Here, the lowest SOC may be mainly explained by wind erosion removing the surface particles in aeolian sandy soil and the process is further accelerated by the poor vegetation cover because of the low soil fertility.

Soil parent materials, climate, and biological, topographical, and other factors have long-term effects on soil formation [3]. Soil type distribution as a means of controlling zonal vegetation is the main factor in the formation of SOC spatial variability [31]. Therefore, soil comprehensively reflects environmental conditions. Under complex environmental conditions, the SOC spatial distribution based on soil type presents obvious differences [3]. Additionally, the changes in soil types may be caused by the changes in vegetation types. Forstner et al. [29] discovered that aeolian sandy soil is the major soil type developed in arid areas with sparse vegetation and little rain; thus, the accumulation of soil organic matter is not obvious in such an area. By contrast, brown soil develops in forest vegetation areas, and the bioaccumulation is much higher in brown soil than in other soil types. Therefore,

ecological construction must be performed in accordance with the principle of adaptation to local conditions, with soil type regarded as an important factor.

### 4.2. Factors Affecting SOC Variability

The results of this study suggest that the three scales of local distribution are similar in the same zone, but more details can be determined as the scale becomes smaller (Figure 3). The differences in SOC at the local scale can be obscured by large-scale continuous processes, such as vegetation and soil morphological differences [32]. As shown in Table 5, the relative importance of each factor changes with scale. Altitude and NDVI are the controlling factors that can explain the SOC content variability at the county scale. SOC significantly changes depending on the topography and vegetation. Characteristic spacing may exist between geologic units dominated by high-activity altitude and NDVI. Similar studies have reported that the correlation of altitude and SOC content is significant, whereas SOC content and slope have no obvious correlation at the county scale [33], and the spatial distribution pattern of SOC content is mainly affected by environmental factors, such as sandy soil textures and terrains [34]. The relationship between altitude and SOC content is close at the county scale, as shown in Figures 1 and 3. Similar distribution patterns were found from south to north in the study area, indicating the transition characteristics from high to low distribution [35]. The change in altitude related to the zonal distribution of vegetation, climate, and soil parent material are important factors affecting the SOC spatial distribution. NDVI is another key predictor controlling the spatial distribution of SOC. Previous studies [32,36] also found that NDVI and altitude are the key factors affecting SOC at the provincial scale, and vegetation structure and density strongly influence soil carbon pool patterns. This trend can be attributed to the key role of vegetation cover in biomass productivity, which is the determinant of litter input [37]. The county scale has greater variability in soil types. Soil and land use types are closely correlated with the distribution of zonal vegetation; thus, their influence on SOC variability can be explained quantitatively and indirectly by NDVI. Similar evidence was obtained by Oueslati et al. [38].

Here, when the scale of the study was changed to a regional level, a new ecological relationship was established based on the analysis of the local environment. In particular, the distribution pattern of SOC was more clearly defined by the slope and NDVI. When vegetation and soil are highly homogeneous, such as at the watershed scale, the content of SOC is largely determined by erosion and sedimentation [39], and more distribution details of SOC can be explained by topography. Thus, local topography plays a key role in the distribution pattern of SOC. Here, the variability of SOC in relation to vegetation gradually became obscured by topography with the reduction of scale, and the topographical factors began to show higher correlation coefficients with SOC at the watershed scale (Table 4). This finding can be explained by erosion usually causing nutrient redistribution and loss, which may directly contribute to the variation in SOC. Thus, topography factors are good determinants of SOC variability [40]. Moreover, soil variations are the result of a comprehensive external effect of topography and land use [41], and the effect also exhibits differences at various scales. The main controlling factors explaining the variability of SOC content may not be directly influenced by soil and vegetation; furthermore, topography variables have a significant stable relationship with SOC in space at the watershed scale. The main threat to the watershed environment is erosion, which tightens the relationship between topography and SOC content. Thus, vegetation construction should be based on erosion intensity. The direct effects of topography on erosion determines the variability of SOC. Similar results have been reported in the literature [42]. There are further reasons for the effect of topography on SOC such as the effects of topography on hydrothermal conditions. The accumulation and decomposition rates of SOC present significant differences under different hydrothermal conditions [28,40,43,44]. Therefore, the changes in topographical factors can reflect the spatial distribution patterns of SOC, which are particularly significant as the study scale becomes smaller.

## 5. Conclusions

The mean levels of SOC at the county, regional, and watershed scales were 7.49, 7.57, and 7.54 g kg$^{-1}$, respectively. Both structural and random factors influenced the spatial variability of SOC with a moderate spatial correlation at the three scales. With the widening of the study scale, the spatial variability caused by the random factors gradually increased.

Altitude and NDVI are the key explanatory variables of SOC content variability at the county scale. NDVI and slope are the key explanatory variables explaining 36.2% of SOC content variability at the regional scale, whereas terrain factors are the most significant factor at the watershed scale. The impact of NDVI on SOC content variability is much greater at the larger study scale than at the smaller scale. Soil types and land use types are the main factors controlling SOC content at the three scales.

**Author Contributions:** Conceptualization, L.L.; Funding acquisition, Y.Y. and F.Q.; Investigation, C.S.; Methodology, X.D.; Resources, P.Z.; Software, Y.L. All authors have read and agreed to the published version of the manuscript.

**Funding:** Funding was provided by the Universities Young Scientific and Technological Talents of the Inner Mongolia Autonomous Region (NJYT22046), the Central Government to Guide Local Scientific and Technological Development (No. 2021ZY0023) and the Project of "Western Young Scholars" in 2021.

**Institutional Review Board Statement:** The study did not require ethical approval.

**Informed Consent Statement:** The study did not involve humans.

**Data Availability Statement:** The study did not report any data.

**Conflicts of Interest:** The authors declare no conflict of interest.

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
