# Peer review of "Multi-Scale Characterization of Spatial Variability of Soil Organic Carbon in a Semiarid Zone in Northern China"

_sustainability, doi:10.3390/su14159390_

Round 1
Reviewer 1 Report
1. P5,2.4 Regional scale sampling, the square area was 2000 km x 2000km, however, it was much larger than sample area in Figure 1 (a) and (b) according the scale in Figure 1.
2. In the 4 resutls, the analysis or discussion including the cited refers should be not in this part.
3. In the 5 discussion, the structural factors and random factors should be discussed in more detail.
Author Response
Response to Reviewer 1 Comments
Point 1: P5,2.4 Regional scale sampling, the square area was 2000 km × 2000km, however, it was much larger than sample area in Figure 1 (a) and (b) according the scale in Figure 1.
Response 1: Thank you for your comment. This was a incorrect description, just a clerical error, the correct square area was 20 km × 20km. I have revised the sentence as“A total of 100 soil samples were collected based on a 2 km × 2 km grid-point system in a 20 km × 20 km square area in the west of the study area.”
Point 2: In the 4 results, the analysis or discussion including the cited refers should be not in this part.
Response 2: We had removed the cited refers from the results.
Point 3: In the 5 discussion, the structural factors and random factors should be discussed in more detail.
Response 3: Thank you for your comment. The structural factors and random factors have be discussed carefully in the Discussion.
Reviewer 2 Report
This study deals with changes in soil organic carbon at different spatial scales in Northern China. Basically, the idea for the study is simple, and tables and figures are clear. However, it seems that some methods should be explained more carefully. For example, I would like to know whether plant roots only were investigated or whether other organic material in soils (e.g., dead microbial biomass) was included. Similarly, more attention could have been paid in preparing the manuscript for evaluation: sometimes values were missing, some figures were in wrong places etc.
Especially, I would encourage the authors to pay attention to explanations relating to statistical analyses. Please, provide explanations relating each analysis step in the same order you provide results in the result section. Furthermore, please, carefully explain each time what the method was, what were the explanatory variables in the models or other analyses.
In Table 5, you present results of a stepwise regression. I am little bit worried about this analysis since we can see in Table 4, that there are rather high correlations between the explanatory variables included in the models. In case highly correlated explanatory variables are included in the same model, the result may be far from true. Thus, please, check these and consider potentially some other way to proceed here.
Somehow the result section included some sentences that belong to Discussion. Thus, present your own results only in Results, and move all speculations or citations to other work to Discussion. I encourage the authors to check language one more time.
Minor comments
Figure 1. Layout: How soil samples were collected? What was the distance between sample locations?
Page 5, line 5. This should be: “…through the study area”.
Page 5, line 6. Please, add space between “15” and “km”. Same in line 9.
Page 4. Subchapters 2.3.-2.6 could be 2.2.1-2.2.4. Thus, it could be easier to follow the story. I would separate the first paragraph from 2.6. to a subchapter 2.2.4.
Page 5, last paragraph. Please, define C0 and C immediately after they have been presented.
Page 5-6. I would combine subchapters 2.7 and 2.8 under title “Statistical analyses”.
Page 6, just before equation 3. This should be: “…with an unknown constant mean is as follows:”
Page 6, line 24. I would not refer to Fig. 4 here since you have not yet presented Fig. 3.
Page 6, chapter 2.8 Statistical analyses. You used stepwise regression here. How could you check that explanatory variables do not correlate too much (they should not do that)? After the second sentence, I would explain what the predictive variables are (i.e., I would move the last two sentences from the end of this paragraph here).
Table 1. This should be: “Descriptive statistics of soil organic carbon contents at different scales…”
Page 7, lines 5-9. Repetition within the sentence. Please, simplify.
Page 7, lines 9-12. This could be in Discussion.
Figure 3. Please, provide Fig. a first, after which Fig. b, and Fig. c is the last one.
Page 7, chapter 4.2, line 7. It sounds curious that you refer to other papers here. Was the result based on your own investigations? If so, please, remove the reference.
Chapter 4.2. This paragraph includes repetition. Please, simplify and do not repeat the results twice. The last sentence belongs to Discussion.
Page 8, last paragraph, line 3. All values presented here do not correspond with those presented in Table 2. Text from line 4 to the end of the paragraph belongs to Discussion.
Page 9, chapter 4.3, lines 1-3. Values are missing here. The last word in the second sentence is wrong. Please, check the language.
Page 9, chapter 4.3, line 6. Maybe you could refer to Fig. 2 here. In the following sentence you say that: “The soil types from south to north were brown, chestnut soil, cinnamon soil and aeolian sandy soil.” However, this does not correspond to that presented in Fig. 2 (according to Fig. 2, the order is brown soil, cinnamon soil, chestnut soil and aeolian sandy soil). The rest of the paragraph is more like text that could be moved to Discussion.
Page 9, last two paragraphs. Why do you refer to Fig. 4b when you talk about watershed scale, and why to Fig. 4c when you talk about regional scale? Please, be more careful. Furthermore, present results only, and move other speculations or explanations to Discussion section.
Page 10, line 1. Replace “Landuse” with “landuse”.
Table 3. Please, replace “Sample number” with “Number of samples”. Furthermore, locate “Soil types” similarly to the left as “Landuse type” is since it is heading for different soil types.
Page 10, lines 7-8. You say here that: “Grassland SOC showed significantly lower values than shrubland and forest land at each scale (p < 0.05) with a lower CV.” Please, check CV values again from Table 3, and follow information gathered from this table if the values presented in the table are correct. The last sentence in the first paragraph in section 4.4 could be moved to Discussion.
Page 10, a paragraph below Table 3. You could remove this sentence from Results since you do not show any results relating to statistical analyses: “Under these conditions, soil erosion and desertification were also important causes for the decline of SOC content.”
Table 4. Please, provide explanation to NDVI here. There is no need to present column “SOC content” with values 1 in this table.
Page 11, second sentence. This does not belong to Results.
Table 5. I do not trust the results in this table. In Table 4, we can see that there are rather high correlations between explanatory variables included in the same models. It is essential to understand that high correlations between explanatory variables may provide unreliable results. Furthermore, why only one p-value per model have been presented? There should be p-value for each explanatory variable to be sure that they are statistically significant. (Please, use the same term for explanatory variables in throughout this manuscript. Now, you have used “predictive” or “independent variable”.) Moreover, provide also SE values for the coefficients.
Page 11-12. The last sentence starting in page 11. Please, check the language.
Page 12, line 3. “We suggested” could be “We also suggest”.
Page 12, last line. “correlations” should be “correlation”.
Page 13, lines 5-8. This sentence in unclear.
Author Response
Response to Reviewer 2 Comments
Point 1:This study deals with changes in soil organic carbon at different spatial scales in Northern China. Basically, the idea for the study is simple, and tables and figures are clear. However, it seems that some methods should be explained more carefully. For example, I would like to know whether plant roots only were investigated or whether other organic material in soils (e.g., dead microbial biomass) was included. Similarly, more attention could have been paid in preparing the manuscript for evaluation: sometimes values were missing, some figures were in wrong places etc.
Response 1:Thank you for your comment. As you know, the plant roots, other organic material and dead microbial biomass were all the important factors which influenced soil organic carbon. however, the smallest study scale was watershed scale in this manuscript, soil types, landuse types and other larger scale factors affect soil organic carbon more significantly in this study(at county regional and watershed scale ). Plant roots and dead microbial biomass were the leading factor at sample plot scale, those factors were not main research objects at larger scale, so we didn’t consider them in this study. In addition, all the missing values and figures have been revised.
Point 2: Especially, I would encourage the authors to pay attention to explanations relating to statistical analyses. Please, provide explanations relating each analysis step in the same order you provide results in the result section. Furthermore, please, carefully explain each time what the method was, what were the explanatory variables in the models or other analyses.
Response 2:I have explained each the method and the explanatory variables in the model and provided explanations relating each analysis step in the “2. Materials and Methods”.
Point 3: In Table 5, you present results of a stepwise regression. I am little bit worried about this analysis since we can see in Table 4, that there are rather high correlations between the explanatory variables included in the models. In case highly correlated explanatory variables are included in the same model, the result may be far from true. Thus, please, check these and consider potentially some other way to proceed here.
Response 3:I have checked the model, the high correlations between the explanatory variables were related to the environment of study area, this was the objective facts. At the same time, all the explanatory variables were the important factors influenced soil organic carbon, so after testing the model by other way, the results did not change.
Point 4: Somehow the result section included some sentences that belong to Discussion. Thus, present your own results only in Results, and move all speculations or citations to other work to Discussion. I encourage the authors to check language one more time.
Response 4: I have moved all speculations or citations to other work to Discussion and checked language carefully.
Minor comments
Figure 1. Layout: How soil samples were collected? What was the distance between sample locations?
Response : The soil samples were collected by using a small soil drill at a soil depth of 0.00-0.20 m , and the distance between each sample location was 3m.
Page 5, line 5. This should be: “…through the study area”.
Response :It has been revised as required.
Page 5, line 6. Please, add space between “15” and “km”. Same in line 9.
Response :It has been revised as required.
Page 4. Subchapters 2.3.-2.6 could be 2.2.1-2.2.4. Thus, it could be easier to follow the story. I would separate the first paragraph from 2.6. to a subchapter 2.2.4.
Response :It has been revised as required.
Page 5, last paragraph. Please, define C0 and C immediately after they have been presented.
Response :It has been revised as required.The variance on a shorter scale than field sampling is found at zero lag distance, which is called nugget effect(C0).
Page 5-6. I would combine subchapters 2.7 and 2.8 under title “Statistical analyses”.
Response :It has been revised as required.
Page 6, just before equation 3. This should be: “…with an unknown constant mean is as follows:”
Response :It has been revised as required.
Page 6, line 24. I would not refer to Fig. 4 here since you have not yet presented Fig. 3.
Response :The places of Fig. 4 and Fig. 3 has been switched.
Page 6, chapter 2.8 Statistical analyses. You used stepwise regression here. How could you check that explanatory variables do not correlate too much (they should not do that)? After the second sentence, I would explain what the predictive variables are (i.e., I would move the last two sentences from the end of this paragraph here).
Response :It has been revised as required.
Table 1. This should be: “Descriptive statistics of soil organic carbon contents at different scales…”
Response :It has been revised as required.
Page 7, lines 5-9. Repetition within the sentence. Please, simplify.
Response :It has been simplified as required.
Page 7, lines 9-12. This could be in Discussion.
Response :It has been revised as required. I have moved the sentence to Discussion.
Figure 3. Please, provide Fig. a first, after which Fig. b, and Fig. c is the last one.
Response :It has been simplified as required, provide Fig. a first.
Page 7, chapter 4.2, line 7. It sounds curious that you refer to other papers here. Was the result based on your own investigations? If so, please, remove
Response :The reference has been removed as required
Chapter 4.2. This paragraph includes repetition. Please, simplify and do not repeat the results twice. The last sentence belongs to Discussion.
Response :The reference has been removed as required
Page 8, last paragraph, line 3. All values presented here do not correspond with those presented in Table 2. Text from line 4 to the end of the paragraph belongs to Discussion.
Response :The has been simplified “The range of SOC content at the three scales were 980, 1 890 to 2 100 m and were increased with the enlargment of the study scale.“.The reference has been removed as required
Page 9, chapter 4.3, lines 1-3. Values are missing here. The last word in the second sentence is wrong. Please, check the language.
Response :It has been revised as required. The values has been added, and all the sentences has been checked.
Page 9, chapter 4.3, line 6. Maybe you could refer to Fig. 2 here. In the following sentence you say that: “The soil types from south to north were brown, chestnut soil, cinnamon soil and aeolian sandy soil.” However, this does not correspond to that presented in Fig. 2 (according to Fig. 2, the order is brown soil, cinnamon soil, chestnut soil and aeolian sandy soil). The rest of the paragraph is more like text that could be moved to Discussion.
Response :The correct order has been revised, and The rest of the paragraph has been moved to Discussion.
Page 9, last two paragraphs. Why do you refer to Fig. 4b when you talk about watershed scale, and why to Fig. 4c when you talk about regional scale? Please, be more careful. Furthermore, present results only, and move other speculations or explanations to Discussion section.
Response : The description has been revised according to the picture.The other speculations and explanations have been moved to Discussion section.
Page 10, line 1. Replace “Landuse” with “landuse”.
Response :It has been revised as required.
Table 3. Please, replace “Sample number” with “Number of samples”. Furthermore, locate “Soil types” similarly to the left as “Landuse type” is since it is heading for different soil types.
Response :It has been revised as required.
Page 10, lines 7-8. You say here that: “Grassland SOC showed significantly lower values than shrubland and forest land at each scale (p < 0.05) with a lower CV.” Please, check CV values again from Table 3, and follow information gathered from this table if the values presented in the table are correct. The last sentence in the first paragraph in section 4.4 could be moved to Discussion.
Response :The CV values presented in the table are correct. The last sentence been moved to Discussion.
Page 10, a paragraph below Table 3. You could remove this sentence from Results since you do not show any results relating to statistical analyses: “Under these conditions, soil erosion and desertification were also important causes for the decline of SOC content.”
Response :This sentence has been removed as required.
Table 4. Please, provide explanation to NDVI here. There is no need to present column “SOC content” with values 1 in this table.
Response :It has been revised as required.
Page 11, second sentence. This does not belong to Results.
Response :The sentence has been removed as required.
Table 5. I do not trust the results in this table. In Table 4, we can see that there are rather high correlations between explanatory variables included in the same models. It is essential to understand that high correlations between explanatory variables may provide unreliable results. Furthermore, why only one p-value per model have been presented? There should be p-value for each explanatory variable to be sure that they are statistically significant. (Please, use the same term for explanatory variables in throughout this manuscript. Now, you have used “predictive” or “independent variable”.) Moreover, provide also SE values for the coefficients.
Response : In Table 4, the correlations between explanatory variables showed environmental characteristics, and the the P-value only showed statistically significant for the model. The same term for explanatory variables has been defined in throughout this manuscript.
Page 11-12. The last sentence starting in page 11. Please, check the language.
Response :All the sentences has been checked.
Page 12, line 3. “We suggested” could be “We also suggest”.
Response :It has been revised as required.
Page 12, last line. “correlations” should be “correlation”.
Response :It has been revised as required.
Page 13, lines 5-8. This sentence in unclear.
Response : All the sentences has been checked.
Round 2
Reviewer 1 Report
I have noticed that a new second author named Yue Yongjie was list, please notes his contribution for this study.
Check the format of references carefully, such as, in refs 10, the words should be in lowercase except the first one in article title.
Author Response
I have noticed that a new second author named Yue Yongjie was list, please notes his contribution for this study.
Response :The new author named Yue Yongjie has been added,the contribution for this study has also been noted.
Check the format of references carefully, such as, in refs 10, the words should be in lowercase except the first one in article title.
Response : All the references have been checked carefully as required.
Reviewer 2 Report
Some of the concerns were considered in the revised version, but unfortunately, there were several ones that had not been corrected although the authors indicated in their response that the corrections had been made:
11) It is still unclear what was included in to determine SOC. The authors explain in their manuscript that: “The visible plant residues left in soil samples were then cleaned for subsequent measurement of SOC content.” Were the plant residues only included in SOC analyses or both plant residues and sieved soil?
22) Chapter 2.3.1, line 4. Please, define “C”.
33) Chapter 2.3.2. Here are statistical methods that still require some modifications (as requested earlier). Line 13: What are control factors? Please, explain this in the text. Lines 16-17: Write land use types to text within brackets. Furthermore, Tukey test was used also for different soil types (see Table 3), but you have not mentioned this at all here. Please, add explanation relating to this and add soil types to the text. Furthermore, you could organize this chapter as follows (to follow the same order as in Results): 1) Tukey test, 2) Correlation analyses, 3) Regression analyses (explain already in the beginning what variables were as predictive variables in the models). Finally, please, rename the heading 2.3.2 as follows: “Other analyses”.
44) It is still unclear whether the regression models are analyzed correctly. Correlations between the explanatory variables are rather high. The authors explain that they have analyzed their data in different ways but no evidence for that has not been provided. They do not explain whether all explanatory variables included (and their coefficients) in the final models are statistically significant or not. Without information on SE and coefficient specific p-values the authors cannot argue that a specific explanatory variable is statistically significant, i.e., that it really influences SOC.
55) Table 3. Move “Soil types” to the left in the first column.
66) A paragraph below Table 3, lines 4-6. Remove this sentence since you have not analyzed these things in your analyses: “Under these conditions, soil erosion and desertification were also important causes for the decline of SOC content.”
77) Table 4. Why did you not remove column “SOC content” although you said that you have removed it? Furthermore, you did not provide any explanation to NDVI although you indicated so in your response.
Other comments:
Language should be checked. There are some very unclear sentences, wrong and missing words, and mistakes here and there.
Please, present Figure 1 before Figure 2.
Equation 2. Why the equation is twice here? “Xi” should be “xi”, similarly as in the equation.
Table 2. Please, define “RSS” here.
Figure 4. The order of subfigures should be: a, b, c and d (Figure 4a to the upper row left, Figure 4b to the upper row right etc.).
Present Figure 3 before Figure 4.
Page 9, last two paragraphs. Provide the last paragraph before the second last paragraph.
Paragraph 4.5, line 4. You present here value 0.5122 but in the table the value is 0.5112. Which one is correct?
Chapter 5.1. Line 2: You should refer to Table 3 here (not to Table 4). Line 6: I do not understand what you mean with “outlier” here.
Author Response
- It is still unclear what was included in to determine SOC. The authors explain in their manuscript that: “The visible plant residues left in soil samples were then cleaned for subsequent measurement of SOC content.” Were the plant residues only included in SOC analyses or both plant residues and sieved soil?
Response : Soil organic carbon only come from soil in this study. The visible plant residues left refers to litters such as dead branches and leaves, so the plant residues could not included in SOC analyses. The sieved soil was the only materials to be used to determine SOC.
- Chapter 2.3.1, line 4. Please, define “C”.
Response :C is structural variance,which represents the variation range of variance caused by spatial autocorrelation in the survey data.
- Chapter 2.3.2. Here are statistical methods that still require some modifications (as requested earlier). Line 13: What are control factors? Please, explain this in the text. Lines 16-17: Write land use types to text within brackets. Furthermore, Tukey test was used also for different soil types (see Table 3), but you have not mentioned this at all here. Please, add explanation relating to this and add soil types to the text. Furthermore, you could organize this chapter as follows (to follow the same order as in Results): 1) Tukey test, 2) Correlation analyses, 3) Regression analyses (explain already in the beginning what variables were as predictive variables in the models). Finally, please, rename the heading 2.3.2 as follows: “Other analyses”.
Response :The control factors, land use types and soil types have been added. In addition, the chapter has been organized as follows:1) Tukey test, 2) Correlation analyses, 3) Regression analyses.
44) It is still unclear whether the regression models are analyzed correctly. Correlations between the explanatory variables are rather high. The authors explain that they have analyzed their data in different ways but no evidence for that has not been provided. They do not explain whether all explanatory variables included (and their coefficients) in the final models are statistically significant or not. Without information on SE and coefficient specific p-values the authors cannot argue that a specific explanatory variable is statistically significant, i.e., that it really influences SOC.
Response :The information on SE and coefficient specific p-values have been provided in Table 5 to explain the statistically significant as required.
55) Table 3. Move “Soil types” to the left in the first column.
Response :It has been revised as required. Both “Soil types” and “Land use types” have been moved in the first column.
- A paragraph below Table 3, lines 4-6. Remove this sentence since you have not analyzed these things in your analyses: “Under these conditions, soil erosion and desertification were also important causes for the decline of SOC content.”
Response : this sentence has been removed as required.
- Table 4. Why did you not remove column “SOC content” although you said that you have removed it? Furthermore, you did not provide any explanation to NDVI although you indicated so in your response.
Response : This column has been removed as required.
I have explained NDVI in 2.2.4 Sampling treatment and data source
NDVI=(LNIR-LR)/(LNIR+LR)
Where LR represents red band reflectance value, and the LNIR represents near infrared band reflectance value. NDVI can reflect the background influence of plant canopy, such as soil, wetland, snow, dead leaves, roughness, and is related to vegetation coverage.
Other comments:
Language should be checked. There are some very unclear sentences, wrong and missing words, and mistakes here and there.
Response :All the sentences has been checked carefully as required.
Please, present Figure 1 before Figure 2.
Response :It has been revised as required.
Equation 2. Why the equation is twice here? “Xi” should be “xi”, similarly as in the equation.
Response :It has been revised as required.
Table 2. Please, define “RSS” here.
Response :It has been defined in Table 2.
Figure 4. The order of subfigures should be: a, b, c and d (Figure 4a to the upper row left, Figure 4b to the upper row right etc.).
Response :It has been revised as required.
Present Figure 3 before Figure 4.
Response :It has been revised as required.
Page 9, last two paragraphs. Provide the last paragraph before the second last paragraph.
Response :It has been revised as required.
Paragraph 4.5, line 4. You present here value 0.5122 but in the table the value is 0.5112. Which one is correct?
Response :It has been revised as required.
Chapter 5.1. Line 2: You should refer to Table 3 here (not to Table 4). Line 6: I do not understand what you mean with “outlier” here.
Response :It has been revised as required. The outlier was abnormal value in data,extremely high or low value would be found in data.
This manuscript is a resubmission of an earlier submission. The following is a list of the peer review reports and author responses from that submission.